# ImageBind3D: Image as Binding Step for Controllable 3D Generation

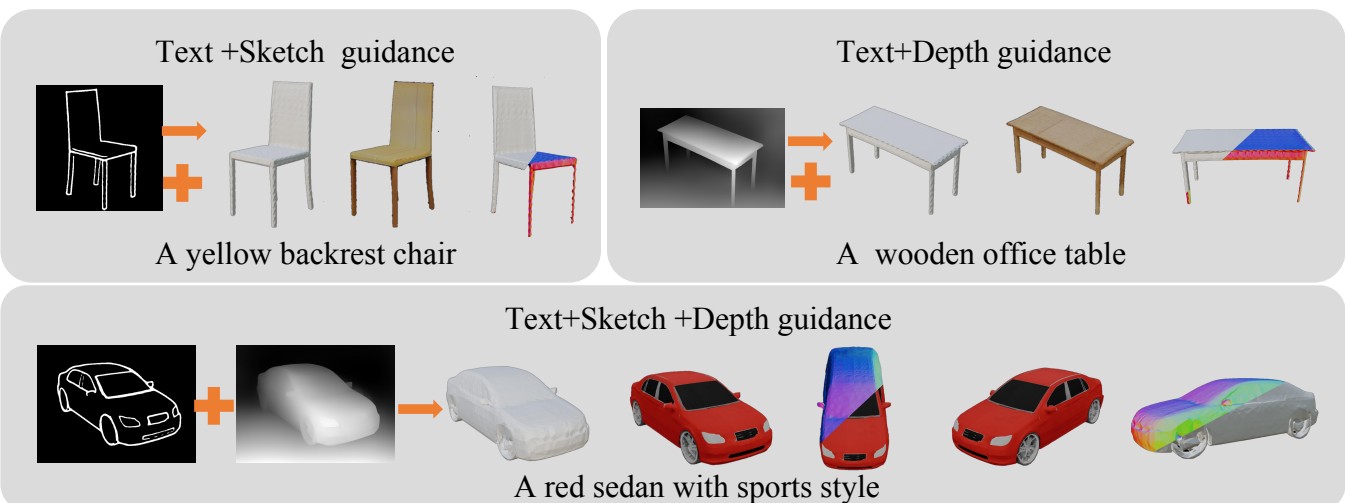

**Figure 1: We propose ImageBind3D, a simple but effective approach that can offer guidance in multiple forms to feed-forward 3D generative models, while not affecting the original network architectures, generation capacity and efficiency. Thanks to ImageBind3D, we can achieve more controllable outcomes, as opposed to the random results generated by GAN-based models or optimization-based techniques (e.g., Get3D and Dreamfusion). Furthermore, ImageBind3D can generate 3D object with composable guidance.**

## ABSTRACT

Recent advancements in 3D generation have garnered considerable interest due to their potential applications. Despite these advancements, the field faces persistent challenges in multi-conditional control, primarily due to the lack of paired datasets and the inherent complexity of 3D structures. To address these challenges, we introduce ImageBind3D, a novel framework for controllable 3D generation that integrates text, hand-drawn sketches, and depth maps to enhance user controllability. Our innovative contribution is the adoption of an inversion-align strategy, facilitating controllable 3D generation without requiring paired datasets. Firstly, utilizing GET3D as a baseline, our method innovates a 3D inversion technique that synchronizes 2D images with 3D shapes within the latent space of 3D GAN. Subsequently, we leverage images as intermediaries to facilitate pseudo-pairing between the shapes and various modalities. Moreover, our multi-modal diffusion model design strategically aligns external control signals with the generative model's latent knowledge, enabling precise and controllable 3D generation. Extensive experiments validate that ImageBind3D surpasses existing state-of-the-art methods in both fidelity and controllability. Additionally, our approach can offer composable guidance for any feed-forward 3D generative models, significantly enhancing their controllability. The code is available at https://imagebind-3d.github.io/imagebind3d/.

## CCS CONCEPTS

• **Computing methodologies → Appearance and texture representations**; **Mesh models**.

## KEYWORDS

3D object generation, Conditional generation, Multimodal diffusion model

*ACM MM, 2024, Melbourne, Australia*
© 2024 Copyright held by the owner/author(s). Publication rights licensed to ACM.
ACM ISBN 978-x-xxxx-xxxx-x/YY/MM
https://doi.org/10.1145/nnnnnnn.nnnnnnn

## 1 INTRODUCTION

High-quality 3D objects are becoming increasingly important in various applications, e.g., metaverse, film special effects, and social platforms. However, the manual creation of 3D assets is very slow and tedious, and requires both specific technical knowledge and refined artistic skills. To accelerate this process, numerous studies have explored the use of generative models[4, 15, 23, 45] for 3D generation, yielding significant advancements. However, these approaches still lack the capability for multi-conditional control.

Existing methods[11, 24, 25, 40] typically requires a large-scale and paired shape data to enable effective training in the field of single-condition-guided 3D generation. Benefiting from 3D supervision, the generated results exhibit commendable geometric fidelity and consistency across different viewpoints. However, these methods exhibit a notable limitation in preserving fine-grained appearance details and ensuring robust controllability. They are constrained to textual inputs and does not have the capacity to incorporate additional guidances, such as sketch or depth map. Yet, the collection and annotation of 3D data present substantial difficulties.

Recently, NeRF[36, 39] and 3D Gaussian Splatting[12, 27, 29, 41], have attracted considerable attention in the field of view synthesis, owing to their remarkable ability to represent complex scenes and produce high-fidelity rendering results. Numerous studies [8, 23, 31, 45] have employed NeRF and GS as 3D representations for text-based 3D generation tasks. Dream Field[23], Dreamfusion[45], and GSgen [8] were introduced to mitigate the constraints posed by limited datasets. By harnessing the capabilities of pretrained vision-language models for guidance, these approaches extract 3D insights from 2D models. These techniques demonstrate excellent performance in generating high-fidelity and coherent 3D objects, meeting various textual prompt provided by the users. Due to the lack of paired textual and shape data, the task of generating 3D shapes from text is highly challenging for the following reasons. On one hand, the 2D diffusion model will introduce biases from the internet dataset into 3D generation. On the other hand, the absence of 3D priors in 2D models gives rise to problems of geometric inconsistency and discontinuity. Additionally, due to the multi-step iterative nature of diffusion models, the generation process often necessitates several tens of minutes.

While these methods can achieve promising generative quality, they notably lack the flexibility in user control capability to accurately guide the generation of 3D objects according to users' specific ideas. Specifically, the absence of multi-conditional control capability results in generated outputs that are usually uncontrolled and unstable. For instance, recent methods like GET3D[15] and Dreamfusion[45] fail to achieve accurate control over generated outputs through the combination of different conditions, e.g., combining text with sketches or depth maps, as shown in Fig.1. In this paper, we try to dig out the control capabilities that 3D generation models have implicitly learned.

Going beyond existing approaches, we introduce a novel 3D generation method named ImageBind3D, which enables multi-conditional 3D generation without the need for matched 3D datasets. Our ImageBind3D methodology adopts an inversion-align two-stage approach that effectively exploits the control capabilities offered by diffusion models, facilitating multi-conditional 3D generation. Inspired by ImageBind[16] and LanguageBind[65], we employ images as an intermediary representation to connect 3D shape with text, sketch, and depth map. In the first stage, employing GET3D as baseline, we design an encoder-based 3D inversion algorithm that aligns images and shapes in latent space, as shown in Figure 2: Stage 1. Next, we extract multi-modal information from images to serve as pseudo-labels for 3D objects. In the second stage, We design a 3D multi-modal diffusion model in the latent space of the 3D GAN and inject additional guiding information into the

diffusion model using decoupled attention, as shown in Figure 2: Stage 2. Utilizing a 3D multi-modal diffusion model, ImageBind3D can generate accurate 3D objects under multiple conditions. We summarize our main contributions as follows:

- We design a 3D multi-modal diffusion model enables accurate control for generating high-quality 3D objects, while also supporting multi-conditional guidance.
- We introduce an encoder-based 3D inversion method to align image and 3D shape in latent space.
- Employing images as an intermediary, we develop a pseudo-label generation strategy between shapes and various modalities, thus eliminating the necessity for matched 3D datasets.

## 2 RELATED WORK

**GAN-based models.** Researchers have explored various methods for generating different 3D representations, including 3D voxel grids [14, 17, 34, 53], clouds [1, 37, 59, 63], implicit models [9, 35, 42, 62], octrees [13, 22], and meshes[3, 15, 21, 30]. However, the primary emphasis of these approaches lies in 3D content generation, with limited attention paid to controllability aspects. Employing semantic or edge maps, Pix2pix3D [10] and SofGAN [6] facilitate new view synthesis, performing admirably for views closely aligned with the input. Yet, when generating views distant from the original conditions, they exhibit degraded quality with rough geometric and lacking in fine details. Closely related to our work, TAPS3D [58] and ISS [32] establish a relationship between the input text and the latent space to achieve text-guided 3D generation. However, they only support text guidance and do not allow for more refined constraints on shape and appearance.

**Diffusion-based generative models.** Diffusion model[20, 54, 55] have recently achieved state-of-the-art performance in multiple generative tasks, such as text-to-image [44, 47, 49, 50], text-to-video[2, 28, 46, 52] and text-to-3D[8, 45, 51]. DreamFusion [45] and SJC [57] employ Neural Radiance Fields (NeRF) to represent 3D structures, subsequently utilize Score Distillation Sampling (SDS) for optimizing the rendering of new perspective images. These methods facilitate zero-shot text-to-3D generation, however, they are constrained by their low-resolution output, slow generation process, over-smoothing, over-saturating, and multi-faceted issues. Concurrently related to our method, HOLODIFFUSION [25] introduces Warp-Conditioned-Embedding [18] as a pseudo-3D representation for 3D diffusion, which is constructed from multi-view features. Subsequently, this representation is rendered into 2D space and further optimized with the aid of 2D diffusion models. It should be noted that their pseudo-3D representation has inspired the 3D representation of our ImageBind3D. Control3D[7] enables the generation of primary views constrained by text and sketches, thus offering a degree of controllability with the SDS strategy. However, it still faces issues such as the inconsistencies of geometry and views, along with slow generation speeds.

## 3 BACKGROUND

**3D genetative model.** Get3D [15] is a novel approach for 3D object generation, capable of producing high-fidelity textured 3D shapes through multi-view supervision. Specifically, GET3D maps noise

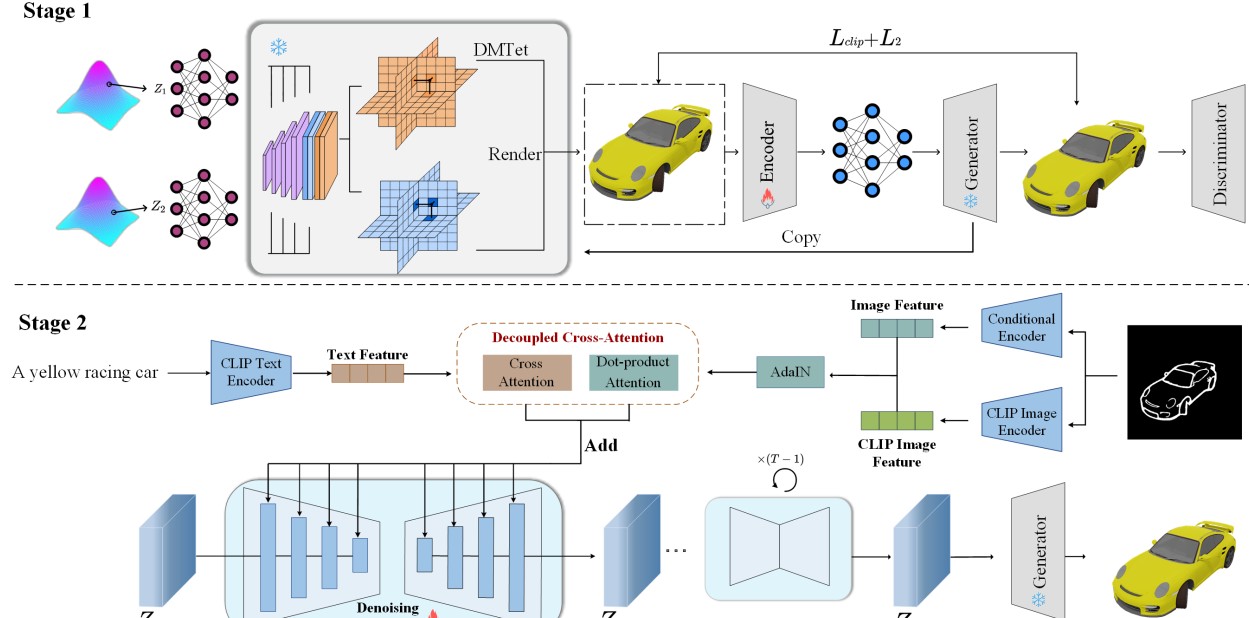

**Figure 2: Our imageBind3D is a two-stage approach for multi-conditional 3D generation. In the first stage, we employ a 3D inversion technique to align images and shapes within the GAN's latent space. Next, we generate pseudo-label centered around the images. In the second stage, we introduce our 3D multi-modal diffusion model for multi-conditional 3D generation.**

vectors $z \in$ N $(0, I)$ to a textured mesh. The generation process includes geometry branch and texture branch. The geometry branch is responsible for the differentiable generation of a surface mesh. Additionally, the texture branch generates a texture field, allowing for color queries to be performed at surface points. Following the design of StyleGAN [26] and PTI [48], they map $z1$ and $z2$ to intermediate latent spaces $w1$ and $w2$. By leveraging the differentiable render, the complete procedure is fully differentiable. The adversarial objective is defined as follows:

$$L(D_x, G) = \mathbb{E}_{z \in N, c \in C}[g(D_x(R(G(z), c)))] \tag{1}$$
$$+ \mathbb{E}_{I_x \in p_x}[g(-D_x(I_x)) \tag{2}$$
$$+ \lambda ||\nabla D_x(I_x)||_2^2], \tag{3}$$

where $g(u) = -\log(1 + \exp(-u))$, $p_x$ represents the distribution of real images, $R$ stands for rendering, and $\lambda$ is a hyperparameter.

**Challenges:** This method enables the rapid generation of high-quality 3D objects; however, it still lacks multi-conditional control capability.

**Diffusion model for Image Synthesis.** Latent Diffusion Models[49] achieved significant advancements in the realm of text-to-image synthesis. T2I-adapter[38] and ControlNet[61] dig out the hidden abilities of T2I models, and then explicitly use them to control the generation, including text, semantic maps, sketches. LDM represents a two-stage diffusion model comprising an autoencoder and a UNet-based denoiser. The optimization process can be expressed by the following formulation:

$$\mathcal{L} = \mathbb{E}_{Z_t, C_{\epsilon,t}}(||\epsilon - \epsilon_\theta(Z_t, C)||_2^2), \tag{4}$$

$Z_t = \sqrt{\bar{\alpha}_t} Z_0 + \sqrt{1 - \bar{\alpha}_t} \epsilon, \epsilon \sim \mathcal{N}(0, I)$ is the noised feature map at step $t$, as a combination of a scaled initial feature map $Z_0$ and scaled noise $\epsilon$, where $\epsilon$ is drawn from a standard normal distribution. C represents the conditional information. $\epsilon_\theta$ is a U-Net-based denoising architecture. Following T iterative steps, the final artifact $\hat{Z}_0$ is propagated into the decoder phase of the autoencoder to perform image generation. They utilize the cross-attention model to incorporate text into the denoising framework, which could be defined as follows:

$$Z' = \text{Attention}(Q, K, V) = \text{softmax}\left(\frac{QK^T}{\sqrt{d}}\right) \cdot V, \tag{5}$$

where $c_t$ is text features, $Q = ZW_q, K = c_t W_k$ and $V = c_t W_v$ represent the query, key, and values from text features. $W_q, W_k$ and $W_v$ represent the weight matrices.

**Challenges:** The primary advantage of this method is its capability for fine-grained control, yet it necessitates a significant investment in computational power and expansive training data. In the process of designing a 3D diffusion model, this issue becomes more evident.

## 4 METHOD

To achieve the best of both worlds, we propose an inversion-align approach named as ImageBind3D, as shown in Figure 2. Our method harnesses the multi-condition guidance capability of diffusion models for Controllable 3D generation. To address computational challenges, we propose denoising in the intermediate latent space of 3D GAN. In response to the absence of paired datasets, we adopt a

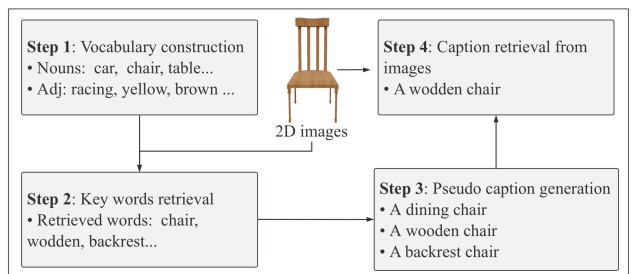

**Figure 3: We demonstrate our pseudo caption generation module, which is inspired by Taps3D[58]. Within this module, there are four distinct steps: Vocabulary construction, key word retrieval, pseudo caption generation, and caption retrieval from images.**

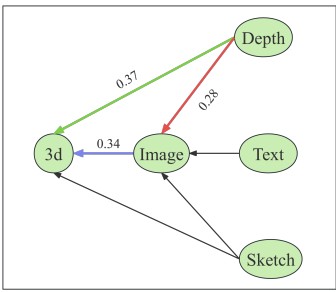

**Figure 4: We calculated the average distance between different modalities using examples from two classes: Car and Chair.**

pseudo-labeling strategy to generate labels for 3D objects. In subsequent sections, we first introduce our 3d inversion method, which aligns images and 3D objects in the latent space (Section 4.1). Subsequently, we discuss our pseudo label generation method, which forms connections between shapes and multiple modalities, with images serving as the central component (Section 4.2). Lastly, we propose our 3D multi-modal diffusion model, which can generate accurate 3D objects according to different input conditions (Section 4.3).

## 4.1 Image-based 3D Inversion

Controllable 3D generation encounters dual challenges: the absence of paired datasets and the substantial computational resources required. To address these two challenges, we design a 3D inversion method. Similar to encoding-decoding architecture of SD[49], we adopt latent codes as 3D representations to simplify computational complexity. Utilizing 3D inversion method, we establishe a mapping relationship between images and 3D representations, which provides us with a benchmark for aligning 3D objects across different modalities. Employing Get3D as the foundational baseline, we develop an encoder-based 3D GAN inversion approach, which adopts an encoder-decoder architecture, as shown in Figure 2: Stage 1. Our encoder architecture consists of VAE encoder and an MLP feature mapping layer. while the decoder follows the generator of Get3D architecture with frozen parameters. We utilize MLP to map image

features from various hierarchical levels to the geometric latent variables $w1$ and appearance latent variables $w2$. By constraining the weights of this generator, our model is able to concentrate on achieving semantic congruence between the input images and the generated 3D objects. Hence, in the training phase, optimization is solely focused on the parameters of VAE and MLP. In the inversion process, we aim to find an intermediate latent variable that to minimize the disparity in reconstruction loss between input images and their 3D renderings. It can be defined as follows:

$$\min_{E} \sum_{i=1}^{N} L(x(i), R(G(E(x(i)), \theta))) \tag{6}$$

where $G(w; \theta)$ is the 3d object generated by Get3D, which parameterized by weights $\theta$, R is a differentiable renderer, $E$ is a VAE encoder.

It is observed that using $L(D_x, G)$ only allows the model to create plausible geometry corresponding to the input image. However, the generated appearance are unnatural and blurry. The primary challenge we face is the discordance between the adversarial loss and our objective of achieving a direct one-to-one mapping during the inversion process. To address this problem, we introduce two additional losses: image similarity loss, and pixel-wise loss. The image similarity loss is defined as follows:

$$L_{\text{CLIP}} = 1 - \cos(E_i(I_x), E_i(I_{gt})) \tag{7}$$

where $E_i$ is the image encoder of CLIP, $I_x$ and $I_{gt}$ present the rendered images and input images. The pixel-wise loss is L2 loss, which signifies the Euclidean norm between the input and rendered images. The overall loss is obtained by blending these three components. It can be defined as follow:

$$L = \lambda_1 L(D_x, G) + \lambda_2 L_2 + \lambda_3 L_{\text{CLIP}} \tag{8}$$

## 4.2 Pseudo Label Generation

The most recent 3D shape generation models are primarily fueled by data. Benefiting from large-scale training data, their performance enhancement is notable. However, when we aim to train a multi-condition 3D generation model, we are confined to a restricted set of multi-modal conditioned 3D objects. To address this challenge, we follow the methodology of [58, 64] and propose generating pseudo-labels for 3D object centered around images. Our pseudo-label strategy is predominantly focused on text descriptions, sketches, and depth maps. For pseudo captions, we adopt a four-step pseudo caption generation method from Taps3D[58], as shown in Figure 3. Initially, we construct a vocabulary using ShapeNet-related[5] nouns and adjectives found in the CLIP vocabulary. Next, we gather multiple words based on the 2D-rendered images. Subsequently, candidate captions are generated using the retrieved words. Lastly, we choose a caption by assessing text-image similarities computed with the CLIP model. Following the paradigm of ControlNet[61], we've incorporated Sketch and Depth estimation models to predict sketches and depth maps.

As shown in Figure 4, through an image-centric pseudo-label strategy, we can efficiently establish connections between multi-modal data.Taking depth, image, and 3D as examples, it is observed that the direct mapping generation of 3D objects results in smaller intervals between modalities. The distance between modalities

is obtained using equation 7. Considering the semantic gap, we opted not to employ a strategy of controlling images with different conditions and then using images to control 3D generation. On the contrary, our objective is to directly establish mapping relationships between different modalities and shapes, without intermediaries.

## 4.3 Multimodal diffusion for controllable 3D generation

Due to the absence of multi-condition guidance in GAN frameworks, we present a 3D multimodal diffusion model for controllable 3D generation, as shown in Figure 2: Stage 2. We have adopted U-Net architecture and decoupled cross-attention module. The multimodal diffusion model is designed to generate diverse latent codes, each corresponding to distinct input conditions. The generated latent codes are inputted as control signals into the generator, thereby enabling controllable 3D generation.

This innovation draws inspiration from the methodologies SD[49] and HoloDiffusion[25]. Specifically, HoloDiffusion leverages multi-view approaches to forge a comprehensive 3D representation suitable for diffusion. In our denoising architecture, $z_t$ is constructed by concatenating geometric latent variables and appearance latent variables, with dimensions of 512*31. Within the latent space, 512×22 dimensions are allocated to present geometric attributes, leaving the remaining 512×9 dimensions to present appearance characteristics. Thanks to our inversion strategy, the computational complexity of our diffusion model is effectively managed at $c * n$.

As illustrated in Equation 5, the original SD model utilizes the cross-attention mechanism to incorporate text into the denoising framework. To achieve multi-conditional control, one straightforward approach is to concatenate the features from disparate conditions and subsequently feed them collectively into the cross-attention layers. However, our findings indicated that this methodology fell short of efficacy. Inspired by Ip-adapter[60], we propose our decoupled cross-attention mechanism, comprising both cross-attention and dot-product attention modules. The output of the cross-attention $Z'$ can be computed by Equation 5. Our dot-product attention module consists of three components: the CLIP encoder, VAE encoder, and AdaIN feature fusion module. These two encoders are employed to extract visual prompt features at semantic and geometric levels. Utilizing these image features, we apply Adaptive Instance Normalization (AdaIN) to normalize two features $c_s$ and $c_g$. It can be defined as follow:

$$\hat{Q}_s = \text{AdaIN}(Q_s, Q_g), \qquad (9)$$

$$\hat{K}_s = \text{AdaIN}(K_s, K_g), \qquad (10)$$

$$\text{AdaIN}(x, y) = \sigma(y)\left(\frac{x - \mu(x)}{\sigma(x)}\right) + \mu(y), \qquad (11)$$

where $x, y$ present CLIP and VAE feature, $\mu, \sigma$ present the mean and standard deviation of features. We concatenate $K_n$ and $\hat{K}_m$, as well as $V_n$ and $V_m$, respectively, to obtain $K_{nm}$ and $V_{nm}$. Our dot-product attention can be defined as:

$$Z'' = \text{Attention}(\hat{Q}_m, K_{nm}^T, V_{nm}) = \text{Softmax}\left(\frac{\hat{Q}_m K_{nm}^T}{\sqrt{d}}\right) V_{nm}, \qquad (12)$$

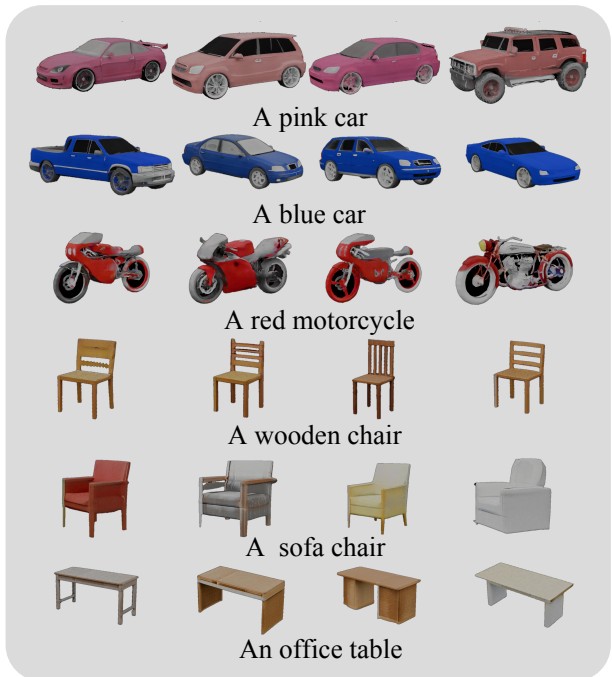

**Figure 5: We demonstrate the generative diversity and textual guidance of our method. Each row employs the text prompt with different samples of random noise as input.**

Where $\hat{Q}_m$, $K_{nm}$ and $V_{nm}$ represent the query, key and value. Subsequently, the output of condition dot-product attention is added to the output of text cross-attention. The decoupled cross-attention is specified as follows:

$$Z^{\text{new}} = \text{Attention}(Q, K, V) + \alpha * \text{Attention}(\hat{Q}_m, \hat{K}_{nm}^T, V_{nm}) \quad (13)$$

$$= \text{softmax}\left(\frac{QK^T}{\sqrt{d}}\right) \cdot V + \alpha * \text{Softmax}\left(\frac{\hat{Q}_m \hat{K}_{nm}^T}{\sqrt{d}}\right) \cdot V_{nm} \quad (14)$$

where $\alpha$ is weight factor, if $\alpha = 0$, the model become the original text-guided diffusion model. Specifically, our additional control signals include sketches or depth maps, as well as combinations of various conditions.

## 5 EXPERIMENT

### 5.1 Implementation Details and Metrics

We conduct training and evaluation on ShapeNet[5]. Our experimental evaluations are performed on four complex geometric categories, including Car, Table, Chair, and Motorbike. We utilize the Get3D model as our 3D generator. In the inversion stage, our inversion experiments are performed with a batch size of 16 and executed on 2 Nvidia 3090Ti-24G GPUs. It costs 15 hours for 3D inversion training. In the alignment stage, we set the batch size to 64. The training process begins with the text-to-3D diffusion model, followed by the training of the multi-modal diffusion model. The training sessions lasted 15 and 6 hours for the respective stages.

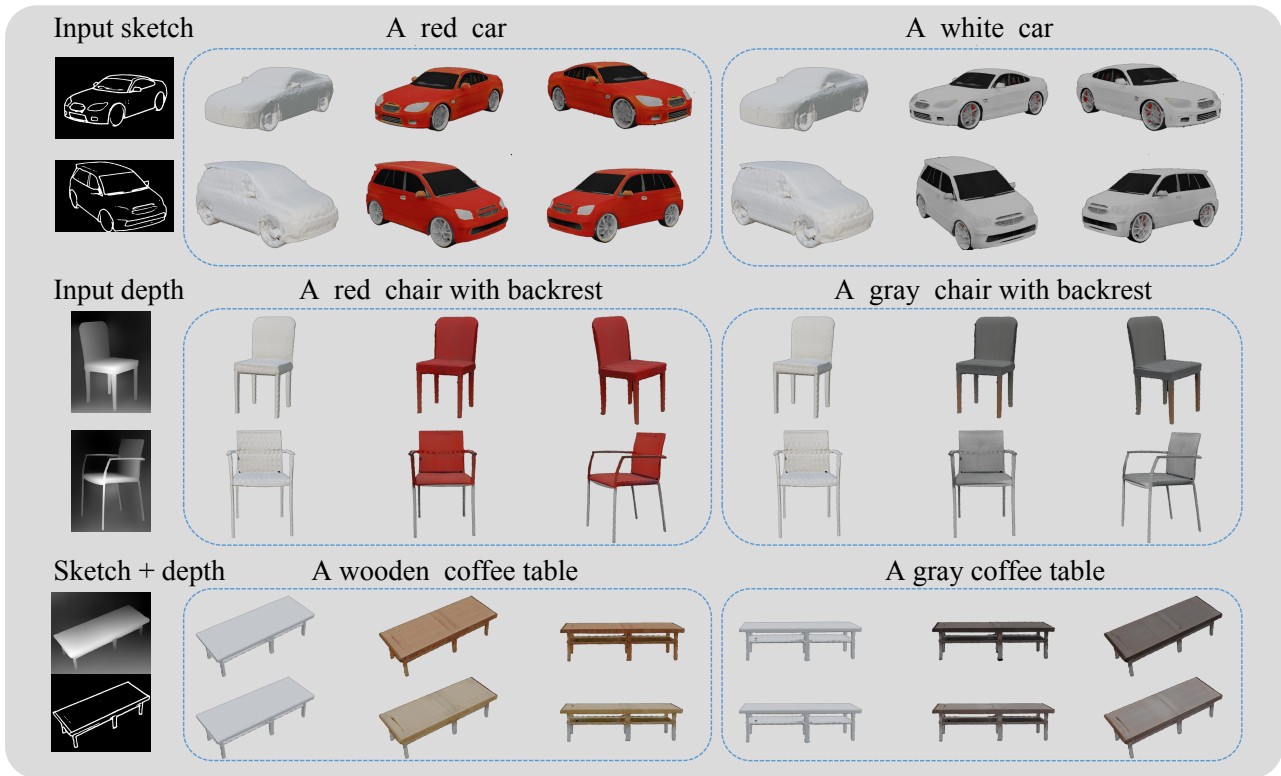

**Figure 6: We show the multi-conditional control capabilities of our method. Each column takes the text prompt and additional conditions as the model input.**

## 5.2 Qualitative Results

In Figure 5, we illustrate our method's generative diversity and textual control efficacy across four object classes. In each row, we sample random noises alongside the provided textual prompts as inputs to generate textured 3D shapes. We observe that the rendered 2D images possess semantic coherence with the provided prompts. Furthermore, the 3D results present substantial diversity in texture and geometric structures, even when generated from the same textual inputs. Additionally, the Motorbike, Table and Chair exhibit more complex geometric structures compared to the Car. However, we still effectively control the texture and geometric details of these challenging categories, demonstrating the robustness of our approach. In Figure 6, we show the multi-conditional control capabilities of our method across different classes. In the initial four lines, we present our ability to generate 3D objects by combining text with sketches or depth maps. Our approach enables the generation of multiple appearances for 3D objects using the same shape prompts and different text inputs. Furthermore, our method can generate diverse 3D objects based on the same text and varying visual prompts. In the last two lines, we present the results of 3D generation based on multiple conditions such as text, sketches, and depth maps. We observed that when we introduce a concept with text and then further refine the object with sketches or depth maps, it leads to more realistic and controllable generation.

This allows ordinary users to rapidly create the 3D objects as they imagine, enhancing the user-friendly design experience.

## 5.3 Comparisons with State-of-the-art Methods

We compared our method with existing works through qualitative and quantitative analyses. To demonstrate the quality of our 3D generation, we compared with existing works using three metrics: FID[19], R-precision[43], and FPD[33]. They are employed individually to assess the quality of rendered 2D images, the distance between textual and image representations, and the geometric fidelity of 3D models.

**Qualitative Comparisons.** Due to the current research methods mainly supporting single-condition guided 3D generation, we performed comparative experiments under different conditions. Our comparative analysis encompasses the following methods: Taps3d[58], ISS[32], and Pix2pix3d[10]. Taps3d and ISS only support text-guided 3D generation, whereas Pix2pix3d enables sketch-guided 3D generation. For fair comparison, we conducted retraining of Pix2pix3d and ISS at a resolution of 1024×1024. In Figure 7, we present the results of qualitative comparisons. We show text-guided, sketch-guided, and combined text and sketch-guided 3D generation separately. We compare across two categories, showcasing the generated results for each category from two different views. For example, with the text "a wooden backrest chair", our Image-Bind3D can generate richer details than taps3d and ISS. Compared

**Figure 7: Our comparative experiments are conducted on two classes: Car and Chair. For each object, we display rendered images from two perspectives. We compare the text-guided 3D generation results of Taps3D[58] and ISS[32] separately, and contrast them with Pix2pix3D[10] for sketch-guided generation results.**

to Pix2pix3D, our method can generate results that better match the sketch description, as shown in the sixth column of Figure 7. It can be observed that our method outperforms Taps3d, ISS and pix2pix3d in generating 3D textured shapes.

**Quantitative Comparisons.** To ensure fairness in comparison, we test on the same dataset using the official codes from GitHub. Table 1 provides quantitative comparisons. Specially, we perform downsampling on our results to match the resolutions of 3DFuse [51]. 3DFuse adopts NeRF for 3D representation and utilizes SDS techniques, thereby imposing limitations on both the resolution and speed of the resultant outputs. In contrast, our backbone Get3D model boasts a larger capacity, enabling higher resolutions. Experimental findings demonstrate that, our approach outperforms 3DFuse in text-guided generation quality for specific categories. Comparing the existing works with ours in Table 1, we can observe that our method outperforms the three state-of-the-art works across all three evaluation metrics.

In Table 2, we compare our method with others in terms of inference speed. Optimization-based techniques, such as DreamFusion[45], 3DFuse[51], and GSGen[8] demand several tens of minutes. Although DreamGaussian[56] reduces optimization time to 2 minutes, the resulting resolution is only 256*256 and the geometric quality is poor. ISS, utilizing optimization strategies for each object, takes approximately 10 minutes. Taps3D, employing direct mapping text feature to latent space, operates in 6.5 seconds. Our approach delivers rendering results in 0.32 seconds and requires 1.09 seconds for mesh generation.

## 5.4 Ablation Study

**3D Inversion Module.** In ablation-1, we removed $L_{clip}$ and $L_2$ during 3D inversion training and conducted experiments following the original inversion-align strategy. All other settings remain unchanged, yet the generated results underperform our original imageBind3D across three evaluation metrics, as shown in table 1. We execute ablative analyses on our methodology under textual

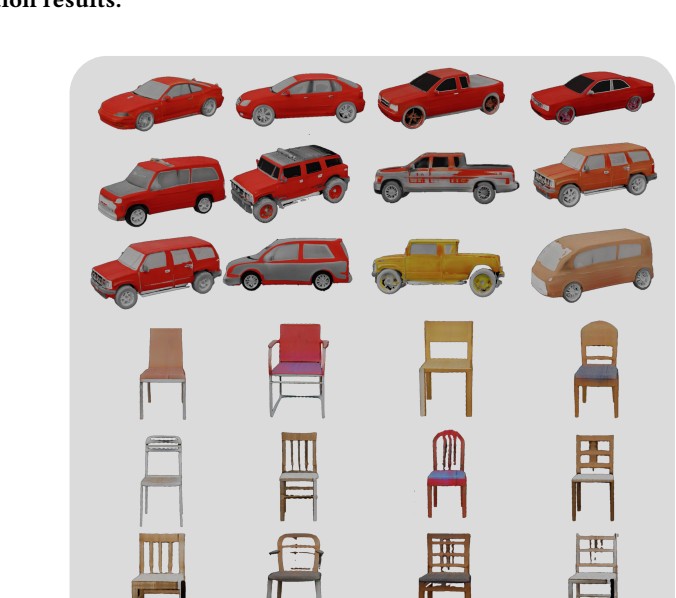

**Figure 8: We perform ablation studies in our method, with two different text prompts "a chair with a backrest" and "a red car". The first row of experimental results corresponds to our complete model, followed by the results of ablation-1 experiment in the second row, and those of the ablation-2 experiment in the third row.**

guidance, encompassing "a red car" and "a chair with a backrest". The experimental results of ablation-1 are visualized in the second and fifth rows of Figure 8.

**Pseudo Label Module.** In ablation-2 of table 1, we directly removed the pseudo label module and utilized a diffusion model to control image directly for 3D generation. The experimental results

**Table 1: Evaluation is performed on the Car and Chair categories using the FID, R-Precision, and FPD metrics. As the resolution generated by 3DFuse is 256×256, we downsample the generated results to for comparison. Ablation-1 and Ablation-2 represent the experimental results of our ablation study.**

| Method | Car | | | Chair | | |
|---|---|---|---|---|---|---|
| | FID (↓) | R-Precision(R=1)(↑) | FPD (↓) | FID (↓) | R-Precision(R=1)(↑) | FPD (↓) |
| 3DFuse[51] | 65.23 | 59.32 ± 2.15 | N/A | 94.75 | 67.55 ± 2.47 | N/A |
| TAPS3D[58] | 34.62 | 62.36 ± 1.93 | 337.67 | 44.83 | 60.19 ± 1.89 | 342.23 |
| ISS[32] | 37.18 | 60.36 ± 2.03 | 364.93 | 44.96 | 58.72 ± 2.23 | 585.79 |
| Ablation-1 | 538.61 | 56.18 ± 1.98 | 1786.42 | 673.18 | 54.07 ± 2.01 | 2487.52 |
| Ablation-2 | 668.35 | 57.13 ± 2.17 | 1875.68 | 797.31 | 55.75 ± 2.04 | 2613.57 |
| Ablation-3 | 35.18 | 61.94 ± 2.06 | N/A | 50.28 | 59.87 ± 1.97 | N/A |
| Ablation-4 | 35.57 | 62.37 ± 2.04 | N/A | 59.69 | 19.52 ± 1.96 | N/A |
| Our-256*256 | **30.46** | **64.05±1.87** | N/A | **40.27** | **64.41±2.05** | N/A |
| Our | **29.04** | **64.57±1.91** | 305.14 | **38.36** | **64.79±1.94** | 322.85 |
| Our+sketch | 28.93 | 64.68 ± 1.90 | N/A | 37.41 | 64.92 ± 1.93 | N/A |
| Our+depth | 28.16 | 64.49 ± 1.93 | N/A | 37.62 | 64.73 ± 1.94 | N/A |

**Table 2: We compare the inference times of various methods across distinct prompts. The inference times for DreamFusion[45] and GSGen[8] were obtained from their papers. For 3DFuse[51], DreamGaussian[56], and ISS[32], we computed the average time from 50 sample sets. For Taps3D[58] and our method, the average time was derived from 500 sample sets.**

| Method | Device | Output | Time |
|---|---|---|---|
| DreamFusion[45] | TPUv4 | Rendering | 90 min |
| 3DFuse[51] | 3090Ti | Rendering | 30 min |
| GSGen[8] | 11G | Mesh | 100 min |
| DreamGaussian[56] | 3090Ti | Mesh | 2 min |
| ISS[32] | 3090Ti-24G | Mesh | 10 min |
| Taps3D[58] | 3090Ti-24G | Mesh | 6.5 sec |
| Ours-text | 3090Ti-24G | Rendering | 0.32 sec |
| Ours-text | 3090Ti-24G | Mesh | 1.07 sec |
| Ours-text+sketch | 3090Ti-24G | Mesh | 1.09 sec |
| Ours-text+depth | 3090Ti-24G | Mesh | 1.08 sec |

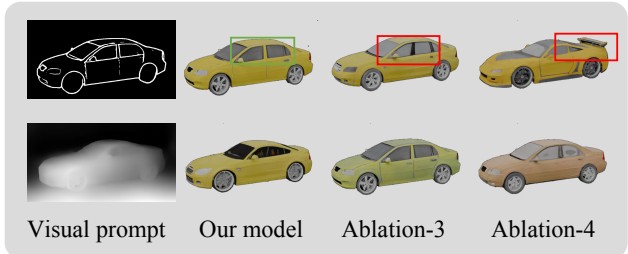

Visual prompt | Our model | Ablation-3 | Ablation-4

**Figure 9: We show 3D results using various fusion mechanisms. All generated outputs are derived from identical text and visual prompts.**

are depicted within the third and sixth rows of Figure 8. Experimental results indicate that pseudo label module plays a significant role in our method. Furthermore, by comparing ablation-1, ablation-2, and the original ImageBind3D, we observe that our inversion-based alignment mechanism contributes more substantially to the overall effectiveness.

**Decoupled Attention Module.** To validate the effectiveness of our decoupled attention module, we conduct ablation-3 and ablation-4. In ablation-3, we adopt the decoupled attention mechanism of IP-Adapter[60]. In ablation-4, we directly add additional information to text, then input them into the cross-attention module. We compared the performance of these two approaches on FID and $R-P$ evaluation metrics, as shown in Table 1. Additionally, qualitative comparisons are illustrated in Figure 9. We merge the textual description "a yellow car" with different visual prompts. In

the first row, the generated result from ablation-3 does not match the sketch conditions for the window part, and in ablation-4, the generated result for the car's tail part is also inconsistent with the input sketch.The experimental results indicate that our decoupled attention module effectively and precisely incorporates different guiding information into 3D generation.

## 6 CONCLUSION

We propose a novel 3D generation framework that enables controllable and high-quality 3D generation with multi-conditional guidance. Initially, we introduce a 3D inversion approach to establish correspondences between images and 3D objects, and then employ the latent codes as 3D representation. Next, we generate pseudo-labels to facilitate model training. Finally, we design a 3D multi-modal diffusion model to control the generation of 3D object. During the inference stage, our method does not require additional optimization steps. Our generation method enables regular users to generate controllable and high-quality 3D objects within acceptable processing times.

**Limitations.** The primary limitation of our method lies in its generation capability, which is constrained by the original generative model. This issue could be addressed by adopting a more extensive and powerful generative model. Besides, we cannot produce different fine-grained details for different object components.

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
