# OpenReview forum: "ImageBind3D: Image as Binding Step for Controllable 3D Generation"
_acmmm.org/ACMMM/2024/Conference — MM2024 Poster_

### Official Review · Reviewer_jaZg · 2024-05-23

**Rating:** 4
**Confidence:** 3

**Summary:**

This paper proposes a method based on GET3D that can generate 3D objects from multimodal information (sketch and depth). Specifically, they first train an encoder to model the mapping relationship between images and the 3D latent space in 3D inversion module. Secondly, they produce pseudo label (depth/sketch->3D latent codes pairs) based on exsiting 3D datasets. Finally, they train a diffusion model to denoise 3D latent codes conditioned on sketch/depth.

**Strengths:**

1.The method is well-motivated and the experimental results are convincing.\
2.The idea of denoising 3D latent codes from GAN with diffusion model in 3D space is interesting.\
3.The cross attention in diffusion model is well-designed.

**Limitations:**

1.Unfair comparison. Most baseline methods in tab.1 and tab.2, such as 3DFuse/dreamfusion/dreamgaussian, focus on open-world objects. However, the proposed method is based on GET3D, which is trained on single category datasets. Do authors take any measures to address the domain gap?\
2.Concern about the quality of inversion results.  It seems that quantitative and qualitative results for inversion module are missing. Besides, the 3D results from inversion are mostly based on the image from side views. Can authors provide some inversion results based on other views(front/back/top/bottom)?\
3.Generalization concern. It seems that most results are generated based on sketch/depth extracted from the datasets. Can authors provide some results based on sketch/depth extracted from in the wild images?

If the authors address my concerns well, I'm glad to increase my rating.

**Suitability:**

3

---

### Official Review · Reviewer_94ZX · 2024-05-24

**Rating:** 4
**Confidence:** 3

**Summary:**

This paper presents ImageBind3D, a new framework for controllable 3D generation conditioned on multiple modalities, such as text, hand-draw sketches and depth maps. To achieve this goal, the framework designed several components. First, the method first developed a 3D inversion technique to inverse an image to latent codes with an encoder-generator method. The latent codes are combined with features from other modalities with a widely-used cross-attention to train a conditional stable diffusion model in the latent space. Experiments show the effectiveness of the proposed method.

**Strengths:**

- A new method to design a multi-modality conditional 3D generator.

- Experiments show the effectiveness of the proposed method.

**Limitations:**

- Does the model train on each category of ShapeNet or the entire dataset?

- How about training on the face dataset and evaluating the multi-modality conditional generation? I understand there could be lots of related work for face generation. It would be nice to evaluate the generalization of the model.

- The paper is a combination of get3D, GAN inversion, and conditional stable diffusion model. It is reasonable but some kind of incremental.

- As discussed in the limitation, the pipeline is limited to the generation quality of get3D like detailed texture.

**Suitability:**

3

---

### Official Review · Reviewer_cpCh · 2024-05-26

**Rating:** 3
**Confidence:** 2

**Summary:**

The authors claim that 3D generation faces persistent challenges in multi-conditional control, primarily due to the lack of paired datasets and the inherent complexity of 3D structures. In light of this, they introduce ImageBind3D, a novel framework for controllable 3D generation that integrates text, hand-drawn sketches, and depth maps to enhance user controllability. They adopt an inversion-align strategy and align external control signals with the generative model’s latent knowledge. Extensive experiments validate that ImageBind3D surpasses existing state-of-the-art methods in both fidelity and controllability.

**Strengths:**

1. The authors introduce an encoder-based 3D inversion method to align image and 3D shape in latent space.
2. Employing images as an intermediary, we develop a pseudo-label generation strategy between shapes and various modalities, thus eliminating the necessity for matched 3D datasets.

**Limitations:**

1. The notations the authors used are not clear and consistent with those in Fig. 2, making it a bit hard to understand the feature fusion strategy.
2. The authors claim that the computation efficacy of their approach. I would expect to see more evidence of the improvement, e.g., the FLOPs comparison.
3. The example cases shown in the paper are some simple objects. It'd be better to generate objects with more complicated shapes and textures for illustration.

Minors:
L404 while should be capitalized
space expected before "Taking..." in L461

**Suitability:**

3

---

### Official Review · Reviewer_LY7G · 2024-06-09

**Rating:** 5
**Confidence:** 4

**Summary:**

This paper introduces a multiple-modal guided 3D generative model for general object. Their methods supports text, hand-drawn sketeches, and depth map as inputs to control 3D generation. The main contributions lie in the proposed pipeline with combinging the 3D gan inversion and latent diffusion models techniques.

**Strengths:**

1) The manuscript demonstrates a smooth flow of ideas, making it easy to follow. The methodology is presented in a clear and accessible manner.

2) Novelty is enough for ACMMM, including  the proposed 3D GAN inversion and multiple-modal guided latent diffusion model.

3) Experiments demonstrates the effectiveness of the proposed method in multiple-modal guided 3D objective generation.

**Limitations:**

1) About details, I'm confused about how to incorporate two-dimensional latent codes as inputs into the U-net of the diffusion model. Could you please provide more clarification or context?

2) Multiple-View results. After reviewing the demo video, I noticed some artifacts in certain results, particularly in the depiction of 3D cars.

3. Typos:

Missing space at line 461.

**Suitability:**

3

---

### Meta-Review · Area_Chair_uLaw · 2024-07-01

**Recommendation:** Accept (Poster)
**Confidence:** 5

**Metareview:**

The paper received (2) weak accepts and (2) borderline accept. All reviewers recognized the multi-modality control that integrates text, hand-drawn sketches, and depth maps for 3D generation. The proposed method is well motivated and novel, the experimental results are sufficient to support the conclusion, and the paper is well written.

Based on their feedback, the decision was made to recommend it for acceptance to ACMMM 2024. We congratulate the authors on their acceptance!